# Study on the Adsorption Behavior between an Imidazolium Ionic Liquid and Na-Montmorillonite

**DOI:** 10.3390/molecules24071396

**Published:** 2019-04-09

**Authors:** Jingjing Pei, Xuesong Xing, Boru Xia, Ziming Wang, Zhihua Luo

**Affiliations:** 1School of Engineering and Technology, China University of Geosciences, Beijing 100083, China; peijj@cugb.edu.cn (J.P.); brxia@cugb.edu.cn (B.X.); wangzm1996@163.com (Z.W.); 2China National Offshore Oil Corporation Research Institute Co., Ltd., Beijing 100028, China; xingxs@cnooc.com.cn; 3Key Laboratory on Deep GeoDrilling Technology, Ministry of Land and Resources, Beijing 100083, China

**Keywords:** ionic liquid, adsorption, Na-montmorillonite, high temperature, interaction

## Abstract

Interactions between 1-butyl-3-methylimidazolium tetrafluoroborate (IL), an ionic liquid, and Na-montmorillonite (Na-MMT) were studied under different kinetic conditions to investigate the adsorption behavior of IL by Na-MMT. The adsorption of IL by Na-MMT was rapid, with a fast rate, reaching a capacity of 0.43 mmol/g, lower than Na-MMT’s cation exchange capacity (CEC) of 0.90 mmol/g. Meanwhile, the highest adsorption rate occurred at the IL concentration of 1000 mg/L. The exchangeable cation of Na-MMT could not be completely substituted by the cation group of IL regardless of the IL concentration. Stoichiometric desorption experiments confirmed that the cation exchange was the dominating adsorption mechanism for the IL adsorption by Na-MMT. The pH value of the solution between 2 and 11 had a negligible effect on the adsorption amount of IL by Na-MMT. The cation group of IL interacted into the interlayer of Na-MMT successfully, resulting in the change in the wettability of Na-MMT. A bilayer formation of the cationic group should occur in the interlayer of the modified Na-MMT and the configuration of IL was dependent on the adsorption amount of IL. Furthermore, the thermal stability of the modified Na-MMT was also dependent on the adsorption amount of IL.

## 1. Introduction

Montmorillonite (MMT) is the best-known group of clay minerals with a three-layer structure. Each crystal platelet consists of an octahedral sheet sandwiched between two-sheet silica tetrahedral. There are predominant substitutions of Al^3+^ for Si^4+^ in the tetrahedral and Mg^2+^ for Al^3+^ in the octahedral, giving montmorillonite negative charge which is balanced by the adsorbed cation (e.g., Na^+^ and Ca^2+^) [1,2]. Of MMT Na-MMT has been widely used in industry, including the treatment of water pollution, drugs, catalysts, nanometer materials, drilling materials, coatings and so on because of its high specific surface, high cation exchange capacity (CEC), common occurrence, and low cost [3,4,5]. For example, Na-MMT is added into a variety of polymer matrix in order to improve the strength and thermal stability of the polymer materials while retaining the lightweight property of polymer materials [6,7]. However, the inorganic Na-MMT cannot disperse well in the polymer matrix. Organic cations can easily exchange with the Na^+^ in the interlayer of Na-MMT and change the wettability of the clay. Hence, organic compounds are used to modify Na-MMT and enhance dispersion and compatibility of the modified Na-MMT in organic phase [8,9,10]. The modification of Na-MMT with conventional organic compounds has been extensively investigated for the past several decades. However, the conventional organic compounds such as quaternary ammonium ions start degradation from 180 °C and have the disadvantage of lower thermal stability, limiting its application [11]. In recent years, treatment of the layered clay with ionic liquids may open a new way in higher-temperature application [12,13].

Ionic liquids (ILs) are organic salts which consist of cations and anions. Different from solid inorganic or organic salts, ILs remain liquid state at room temperature. Due to their unique properties such as “zero” vapor pressure, high thermal stability, good solvent properties with organic, inorganic materials, ILs have attracted significant interest and are considered as an alternative to conventional organic compounds for a wide range of industrial applications [14,15,16,17]. One important property of ILs is its high thermal stability. Most of ILs remain stable in a liquid state at 300 °C and some of them remain stable even up to 400 °C. ILs have been used to modify Na-MMT, which can overcome the shortcomings of conventional organic compounds and improve the thermal stability of Na-MMT with polymer materials [18,19]. Livi et al. reported that Na-MMT modified with an imidazolium ionic liquid has excellent thermal stability compared with conventionally treated Na-MMTs [18]. Laurence et al. used a phosphonium ionic liquid to improve the thermal stability of MMT up to 330 °C [19]. In our previous work, we investigated the influence of an imidazolium ionic liquid on the rheological and filtration performance of water-based drilling fluids at high temperatures. The ionic liquid (abbreviated as IL) with a very low concentration was directly added into drilling fluids and it can improve the performance of drilling fluids at high temperatures up to 240 °C. Hence the ionic liquid has a potential application in deep drilling of petroleum, gas or geothermy [20]. However, these reports mainly focus on the application of the IL-interacted Na-MMT at high temperature rather than the microscopic mechanism of the IL by Na-MMT through the adsorption behavior.

This study mainly focuses on the adsorption behavior of the IL by Na-MMT, including the adsorption kinetics, the adsorption isotherm, the desorption of exchangeable cations and microscopic structure of the interacted IL in the interlayer, in order to further understand the interaction mechanism between the IL and Na-MMT.

## 2. Results and Discussion

### 2.1. IL Adsorption Kinetics

The study was aimed to investigate the needed time to reach adsorption equilibrium so that the adsorption equilibrium can be established at a fixed time in a further study. The adsorption of IL by Na-MMT was quick, maybe instantaneous (Figure 1a), similar to some cation adsorption [21]. The maximal adsorbed amount of 0.436 mmol/g at 30 min, had basically no change from the adsorbed amounts of 0.429 mmol/g at 5 min. The rapid adsorption of IL by Na-MMT suggested the ionic liquid had a great affinity for Na-MMT surfaces. Na-MMT provided a great advantage due to its high cation exchange capacity (mainly Na^+^) for the cationic group of IL. The adsorption equilibrium would be established at least after 1 h of contact with Na-MMT in the further study.

Several kinetic models were fitted to the experimental data and only the pseudo-second-order model can describe well the adsorption of IL by Na-MMT. The pseudo-second-order model has the form:(1)$$\frac{t}{q_{t}}=\frac{1}{k_{s}q_{e}{}^{2}}+\frac{1}{q_{e}}t.$$

The fitted plot was shown in Figure 1b and we can see that the adsorption of IL by Na-MMT fitted very well the model with a coefficient of determination R^2^ 1.0000. The plot of t/q_t_ vs.t could be used to judge the surface heterogeneity of Na-MMT, and the plot of t/q_t_ vs.t would not be a straight line or divided into two segments if the surface of Na-MMT was heterogeneous [22]. In this study, the plot is a perfectly straight line, showing that the adsorption site of the clay mineral was homogeneous, similar to reported literature [22]. For adsorption of IL by Na-MMT, q_e(cal)_, the maximal adsorbed amount at equilibrium time, of 0.43 mmol/g was obtained by the pseudo-second-order model, equal to q_e(exp)_, the experimental value (Table 1).

### 2.2. IL Adsorption Isotherm

Based on the kinetics studies, the adsorption time was fixed at 1 h by which equilibrium would be established to further study. Adsorption amounts versus initial concentrations of IL and adsorption rate versus initial concentration of IL are shown in Figure 2. The adsorption rate (r) is obtained by the following equation:r = (q_e_ M)/(C_0_V) ∗ 100%(2) where q_e_ is the adsorption amount of IL by Na-MMT at equilibrium (mmol/g), M is the weight of adsorbent (g), C_0_ is the initial concentration of IL (mmol/L) and V is the volume of the IL solution.

For the adsorption of IL by Na-MMT, the adsorption amount at equilibrium (q_e_) firstly increased with the initial concentration of IL (Figure 2a). The value of q_e_ is 0.43 mmol/g at the initial concentration of 1000 mg/L, similar to the result of the kinetics study. The q_e_ reached the maximum value of 0.70 mmol/g at the initial concentration of 2000 mg/L, and then the value of q_e_ has a slight decrease at the initial concentration of 5000 mg/L. Meanwhile, the highest adsorption rate (r) of 98.0% was observed at the initial concentration of 1000 mg/L (Figure 2b). But when the initial concentration of IL increased to 2000 and 5000 mg/L, the adsorption rate (r) decreased to 78.0% and 31.0%, respectively. Meanwhile, the CEC value of 0.9 mmol/g for Na-MMT was larger than the maximum value of 0.70 mmol/g at the initial concentration of 2000 mg/L. These results indicated that the cation group of IL could not substitute the whole exchangeable cation of Na-MMT even if the concentration of IL was increased. The adsorption capacity of IL by Na-MMT not only depends on the initial concentration of IL but also on the cation exchangeable capacity of Na-MMT, which has been reported in other literature [23]. For the adsorption of IL by Na-MMT, the adsorption rate would not be 100% and the highest adsorption rate of IL by the clay mineral occurs at 1000 mg/L. This conclusion also provides the basis for the application of IL.

The experimental data were fitted to Langmuir and Freundlich isotherm model, respectively. The Freundlich fit with a coefficient of determination R^2^ 0.31 resulted in a big discrepancy between the experimental and calculated data, indicating the adsorption of IL by Na-MMT cannot follow the Freundlich isotherm model. Compared to the Freundlich fit, the better fit to the Langmuir model with a higher coefficient of determination R^2^ 0.78 but the adsorption of IL by Na-MMT cannot follow Langmuir model well, suggested a surface-limited adsorption for the adsorption. The adsorption of other cations by Na-MMT, such as chlorpheniramine and tetracycline also were described by the Langmuir model, similar to the adsorption of IL by Na-MMT [5,24]. The configuration of IL in the Na-MMT interlayer may be not monolayer because Langmuir isotherm model is established on the hypothesis of monolayer adsorption of adsorbate by the adsorbent.

Langmuir isotherm can be transformed into a linear form:(3)$$\frac{C_{e}}{q}=\frac{1}{q_{m}}C_{e}+\frac{1}{q_{m}K_{b}}$$
where C_e_ is the concentration of IL solution (mmol/L), q is the adsorbed amounts of IL at equilibrium (mmol/g), q_m_ is the maximum adsorption amount (mmol/g), and K_b_ is the Langmuir coefficient (L/mmol). So that K_b_ and q_m_ can be obtained by the linear regression in Table 2.

The fitted q_m_ and K_b_ values were 0.83 mmol/g and 4.60 L/mmol, respectively, comparable to chlorpheniramine adsorption of 0.64 mmol/g and 10 L/mmol by a similar Na-MMT [5,17]. The K_b_ can be used to calculate the adsorption free energy (ΔG^°^) by the following equation [7]:(4)$$\Delta G^{o}{}_{ads}=-RT\ln(K_{b}\times 55.5).$$

The calculated value of ΔG^°^ for the adsorption of IL by Na-MMT were −13.7 kJ/mol, close to the value of −15.8 kJ/mol for chlorpheniramine adsorption by Na-MMT, higher than that of −6.2 kJ/mol for chlorpheniramine adsorption by talc, further showing that the ionic liquid has a great affinity for Na-MMT surfaces. Na-MMT provides a great advantage due to its high cation exchange capacity for the cationic group of the IL.

### 2.3. Cation Desorption Accompanying IL Adsorption

Figure 3 shows the desorbed cations amounts vs. the adsorption amount of IL. It can be seen that the total desorbed amount increases with the adsorption amount of IL. There was a better linear correlation between the adsorbed amount of IL and total desorbed amounts, indicating that cation exchange adsorption is mainly responsible for the IL adsorption by Na-MMT, similar to the adsorption of other organic compounds on Na-MMT [17,18]. The total desorbed amounts of exchangeable cations are about 0.72 mmol/g, lower than the CEC value of the raw Na-MMT, further indicating that the cation group of IL could not substitute the total cation of the clay mineral. Meanwhile, most of the desorbed cations were Na^+^, confirming that the montmorillonite was the Na-montmorillonite in Na form.

### 2.4. Influence of Solution pH on IL Adsorption

Figure 4 shows the adsorption amounts of IL by Na-MMT at different solution pH. It was obvious that the adsorption amounts of IL by Na-MMT basically remained at about 0.43 mmol/g as the solution pH was increased from 2 to 11. The adsorbed amounts of IL remained relatively stable in wide pH ranges tested, indicating that solution pH had much less effect on the adsorption amounts of IL by Na-MMT, similar to the adsorption of some organic compounds by Na-MMT [17,18]. Thus, the steady adsorption amounts of IL between pH 2 and 11 could be attributed to pH-independent cation concentration of IL between pH 2 and 11. However, when solution pH was increased from 11 to 13, the adsorption amounts of IL greatly decreased from 0.43 to 0.22 mmol/g. When the pH value of the dispersion increased to 13, the strong base solution may result in lower zeta potential of Na-MMT. Then, fewer cations may insert into the interlayer of Na-MMT with lower zeta potential. That is to say, less amount of IL may be adsorbed by Na-MMT in solution with high pH value.

### 2.5. XRD

X-ray diffraction (XRD) was used to measure d_001_-spacing values of Na-MMT, which is dependent on the adsorbed exchangeable cation. Figure 5 shows the XRD patterns of modified Na-MMTs with different concentrations of IL, modified Na-MMTs at different solution pH and a different time. The raw Na-MMT was also measured for comparison. The d_001_-spacing of raw Na-MMT was 12.3 Å (as shown in Figure 5a), similar to the value as reported in many works of literature [23,24]. A progressive shift to 14.0 Å could be observed at IL concentration of 100 mg/L. However, when the concentration of IL increased to 1000 mg/L, the d_001_-spacing of the Na-MMT decreased to 12.7 Å. When the concentrations of IL further increased to 2000 mg/L and 5000 mg/L, the d_001_-spacing of Na-MMTs remained stable of 13.3 Å. The results show that the d_001_-spacing of Na-MMT was not only dependent on the adsorption amount of IL but the arrangement of the interacting cationic group in the interlayer of Na-MMT. The different d_001_-spacing of Na-MMTs adsorbed with different amounts of IL may be induced by the different configuration of the interacting cationic group in the interlayer of Na-MMTs. The d_001_-spacing of raw Na-MMT with one layer of water is 12.3 Å. The interlayer distance can be obtained by subtracting the thickness of the dehydrated Na-MMT layer (9.6 Å) [22] from the observed d-spacing of the modified Na-MMT with IL. But the height of the imidazole group with alkyl chains was about 3.3 Å [25]. Maybe the cationic group of the IL adopts vertical orientation which leads to larger d_001_-spacing of the modified Na-MMT with a lower concentration of the IL (100 mg/L). When increasing the concentration of the IL, the cationic group of the IL adopts lying parallel orientation, which leads to smaller d_001_-spacing of the modified Na-MMT, similar to that reported in the literature [25].

The d_001_-spacing of the modified Na-MMTs as the initial concentration of IL was 1000 mg/L varied with different time and different pH (as shown in Figure 5b and Figure 5c), suggesting that the d_001_-spacing of Na-MMT should be not only dependent on the adsorption amount of IL but also the adsorption time and the pH value of the solution. The adsorption behavior is dynamic during the adsorption process even if the adsorption reaches equilibrium. The orientation adopted by the adsorbed cation group of IL at a near constant adsorption amount of 98.0 mg/g may vary with the different adsorption time or different solution pH. The adsorbed cations of IL may adopt a tilt orientation at one time or solution pH and at another time or solution pH it may adopt vertical orientation in the interlayer of Na-MMT, which is different from that reported in other literature [5].

### 2.6. FT-IR

The chemical composition of materials can be measured qualitatively through fourier transform infrared (FT-IR) analyses. Figure 6 shows the FT-IR spectra of IL, raw Na-MMT and Na-MMTs modified with IL. The adsorption peaks of Na-MMT in the range of 3440–3624 cm^−1^ and 1640 cm^−1^ were ascribed to O-H stretching and bending vibration band, respectively, which was also reported in other literature [23,24]. Compared with raw Na-MMT, the characteristic absorption peaks of IL could be observed in the FTIR spectrum of the modified Na-MMTs. For the modified Na-MMTs, the characteristic peaks at 3320 to 2850 cm^-1^ correspond to C-H absorptions of the alkyl group of IL. And the characteristic peaks of the imidazolium group could be observed at 1635 and 1580 cm^−1^, which is resulted from the C=N and C=C frame vibration. These results indicated that the cationic group of IL had successfully intercalated into the Na-MMT interlayer, similar to the exchangeable cation adsorption of organic compounds [26,27,28].

### 2.7. TGA

Figure 7 shows the thermogravimetric analyzer (TGA) patterns of IL, raw Na-MMT and the modified Na-MMTs with different concentrations of IL. As shown in Figure 7a, raw Na-MMT lost its weight at approximately 100 °C, due to the loss of free water in the interlayer of Na-MMT. When the concentrations of IL were 2000 mg/L and 5000 mg/L, the thermal degradation temperatures of the corresponding Na-MMTs adsorbed with 0.70 mmol/g and 0.686 mmol/g IL were about 400 °C and 350 °C, respectively. However, when the concentration of IL decreases to 1000 mg/L, the thermal degradation temperature of the corresponding Na-MMT adsorbed with 0.43 mmol/g IL was about 500 °C. When the concentration of IL further decreased, the thermal degradation temperature of the corresponding Na-MMT adsorbed with a lower amount of IL was beyond 500 °C. These results confirmed that the thermal stability of Na-MMTs adsorbed with lower IL amounts were better than that of Na-MMTs with higher IL amounts, probably due to chemical adsorption of Na-MMTs with lower IL amounts and physical adsorption of Na-MMTs with higher IL amounts [21,22,23]. Meanwhile, Na-MMT adsorbed with IL had better thermal stability than Na-MMT adsorbed with a quaternary amine, the degradation temperature of the latter was reported to be about 180 °C [11].

### 2.8. Contact Angle Measurement

The contact angle is used to characterize the wettability of a material surface. Higher contact angles indicate that the material is more lipophilic than hydrophilic. Figure 8 showed the contact angles of raw Na-MMT and modified Na-MMT with different concentrations of IL. The contact angle of raw Na-MMT was 20°, equal to the reported value. The Na-MMT modified with 1000 mg/L of IL showed a slightly higher contact angle (ca.40°), suggesting that the modified Na-MMT tend to be more lipophilic. When IL concentrations increased to 2000 mg/L and 5000 mg/L, the contact angles of the modified Na-MMT were 45° and 50°, respectively, much higher than that of raw Na-MMT. These results show that the insertion of IL into the interlayer of Na-MMT could change the wettability of Na-MMT and the modified Na-MMT tends to be more lipophilic. The cation group of IL with alkyl chain exchanged with the Na^+^ of Na-MMT, which enhanced the lipophilicity of the Na-MMT.

### 2.9. Atomic Arrangement of IL in the Interlayer of Na-MMT

Previous studies showed that the configuration of an intercalated cation in the Na-MMT interlayer was largely dependent on the concentration of intercalated molecules [25]. In this study, the d_001_-spacing of raw Na-MMT with one layer of water is 12.3 Å. The interlayer distance (4.2 Å) can be obtained by subtracting the thickness of dehydrated Na-MMT layer (9.6 Å) [22] from the observed d-spacing of the Na-MMT modified with IL of 100 mg/L (14.0 Å). However, the ionic height of imidazole group with alkyl chain was about 3.3 Å [25]. The result suggested that a bilayer formation of the cationic group should occur in the interlayer of the modified Na-MMT. The cationic group of IL may adopt a special orientation in the interlayer and its orientation may be parallel or tilting.

A molecular simulation was performed to determine the atomic arrangement of IL and the locations of two N atoms (Figure 9). The results show that the intercalated cation groups of IL adopted a bilayer formation, which further confirmed the previous analysis. But one layer of the intercalated cation groups adopted parallel orientation and another adopted vertical state in the interlayer of Na-MMT (Figure 9a). The distance from the basal plane of interlayer space to the N on the shorter alkyl chain is 2–3 Å or 14–16 Å, and the distance from the basal plane of interlayer space to another N on the longer alkyl chain is 6–8 Å or 10–12Å. The two N atoms in the imidazole ring can be found to have two positions in the interlayer of Na-MMT: one is in the middle of the interlayer space while the other is closed to the basal plane of the interlayer space.

## 3. Experimental Work

1-butyl-3-methylimidazolium tetrafluoroborate (CAS#: 174501-65-6), abbreviated as IL, was provided by Chenjie Chemical Co. Ltd., Shanghai China. It has a molecular weight of 226.02 g/mol and its purity was 99%. Its water solubility is 2–4 g/mL and its pH is 5 at 20 °C. Its formula is shown in Figure 10. Na-montmorillonite (Na-MMT) was obtained from Zhejiang Fenghong new material company, Huzhou, Zhejiang China. The density and CEC of Na-MMT are 2.70 g/cm^3^ and 0.9 mmol/g, respectively. It is composed of 95% smectite, 4.0% quartz and 1% mica+gypsum+feldspar and its formulas is written as Ca_0.12_ Na_0.32_K_0.05_)[Al_3.01_Fe(Ⅲ)_0.41_Mg_0.54_][S_i7.98_Al_0.02_]O_20_(OH)_4_.

Na-MMTs modified with different initial concentrations of IL varying from 100 to 5000 mg/L were prepared for the adsorption isotherm study. Na-MMT (0.2 g) was dispersed in 20 mL of IL solution and the suspensions were shaken for 24 h at 30 °C. After being centrifuged at 7600 rpm for 30 min, the suspensions were filtered through 0.45 μm syringe filters, and then the filter liquor was analyzed for the concentrations of IL and the desorbed exchangeable cations. Each precipitate was washed with distilled water at least three times until no IL could be detected in the supernatant by a UV-Visible spectrophotometer (Model T6, China). The clay samples were obtained by drying the precipitates at 105 °C for 24 h in an oven before measurement. The IL concentration was fixed at 1000 mg/L for the study of adsorption kinetics and pH effects on the adsorption of IL by Na-MMT. The pH value of the Na-MMT/IL dispersion was adjusted to about 2, 3, 5, 7, 9, 11 and 13 with HCl or KOH.

The IL concentrations were measured with UV-Vis spectrophotometer at the wavelength of 211 nm [29], corresponding to its maximum absorbance wavelength. The calibration curve was obtained by measuring the absorbance values of the IL solution with varying concentration (10, 20, 30, 40, 50 and 60 mg/L) and the regression coefficient was greater than 0.999 (Figure 11). The adsorbed amount of IL can be calculated by the difference between the final and initial IL concentration. The equation is as follows:(5)$$q_{t}=\frac{(C_{0}-C_{t})V}{M}$$
(6)$$q_{e}=\frac{(C_{0}-C_{e})V}{M}$$ where q_t_ and q_e_ are the corresponding adsorbed amount of IL by Na-MMT at t and equilibrium time, respectively (mmol/g). C_t_ and C_e_ are the corresponding concentration of the IL at t and equilibrium time, respectively (mmol/L). C_0_ is the initial concentration of the IL (mmol/L), V is the volume of IL solution (L), M is the weight of adsorbent (g).

Each test was conducted three times, and the error is less than 5%.

The desorbed exchangeable cations were measured by atomic adsorption on a AAnalyst-100 Atomic Absorption Spectrometer (Perkin Elmer Corporation, Waltham, MA, USA). Seven standard solutions with concentrations from 0.2 to 3.0 mg/L for K^+^, Na^+^, and Mg^2+^, and from 1.0 to 25.0 mg/L for Ca^2+^ were used to make the calibration curve. The detection limit was 0.006, 0.01, 0.06, and 0.4 mg/L for Na^+^, K^+^, Mg^2+^ and Ca^2+^ with detection wavelengths at 589.0, 766.5, 285.2, and 422.7 nm, respectively.

The clay samples were scanned using a Ultima X-ray diffractometer (XRD) (Rigaku Corporation,, Tokyo, Japan) with Cu-Kα radiation (40 kV, 100 mA). The XRD patterns were collected from 2° to 50° at a 2 °/min scanning rate. Fourier transform infrared (FT-IR) spectra of raw Na-MMT, IL and the modified Na-MMT were obtained by a NEXUS-650 FT-IR spectrometer (Nicolet corporation, Madison, WI, USA). The thermal stability of the clay samples was measured by the SDT Q600 thermogravimetric analyzer (TGA) (TA instruments, New Castle, DE, USA) from room temperature to 600 °C, operating at a heating rate of 10 °C/min in an air atmosphere. The contact angle was tested on a JC 200D contact angle instrument (shanghai zhongchen Digital Technology Co.,LTD, Shanghai, China) with deionized water. The testing samples were prepared according to a method in the literature [30]. A molecular simulation was performed to investigate the atomic arrangement of IL in the interlayer of Na-MMT via the module Forcite of materials studio software 5.0.

## 4. Conclusions

The adsorption of IL by Na-MMT was rapid at a fast rate. The adsorption kinetics of IL by Na-MMT followed the pseudo-second-order model very well, and its adsorption equilibrium amount was 0.43 mmol/g, lower than the CEC value of 0.90 mmol/g. The adsorption of IL by Na-MMT reached the maximum adsorption amount of 0.70 mmol/g at the IL concentration of 2000 mg/L, but the highest adsorption rate occurred when the IL concentration was 1000 mg/L. Cation exchange was the dominating adsorption mechanism and the cation group of IL could not completely substitute the exchangeable cation of Na-MMT. XRD and FTIR analysis indicated that the cation group of IL interacted into the interlayer of Na-MMT, which changed the wettability and improved the thermal stability of Na-MMT. A bilayer formation of the cationic group should occur in the interlayer of the modified Na-MMT, the configuration of IL and the thermal stability of the modified Na-MMTs were dependent on the IL concentrations.

## Figures and Tables

**Figure 1 molecules-24-01396-f001:**
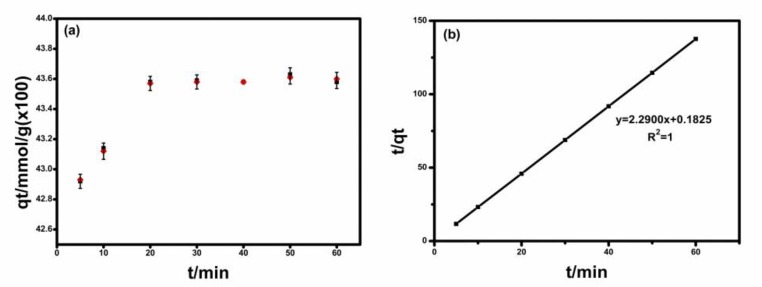
(**a**) Influence of time on the adsorbed amount of the ionic liquid (IL)IL. (**b**) t/q_t_ at different times and the fitted plot.

**Figure 2 molecules-24-01396-f002:**
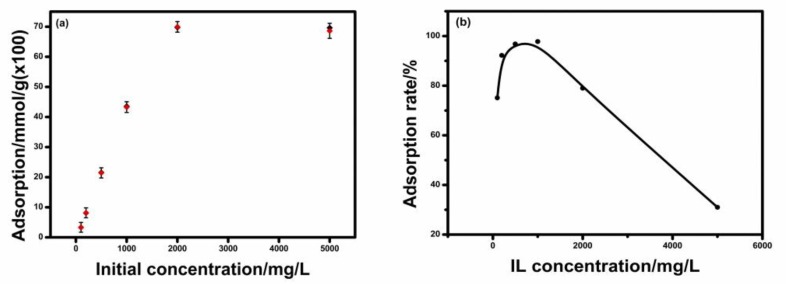
(**a**) Adsorption amount of IL versus initial concentrations of IL; (**b**) adsorption rate versus initial concentration of IL.

**Figure 3 molecules-24-01396-f003:**
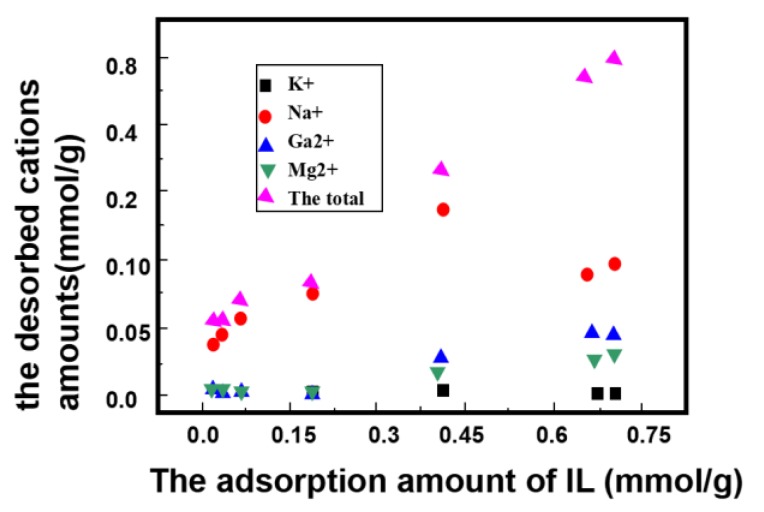
The desorbed amount of cations versus the adsorption amount of IL.

**Figure 4 molecules-24-01396-f004:**
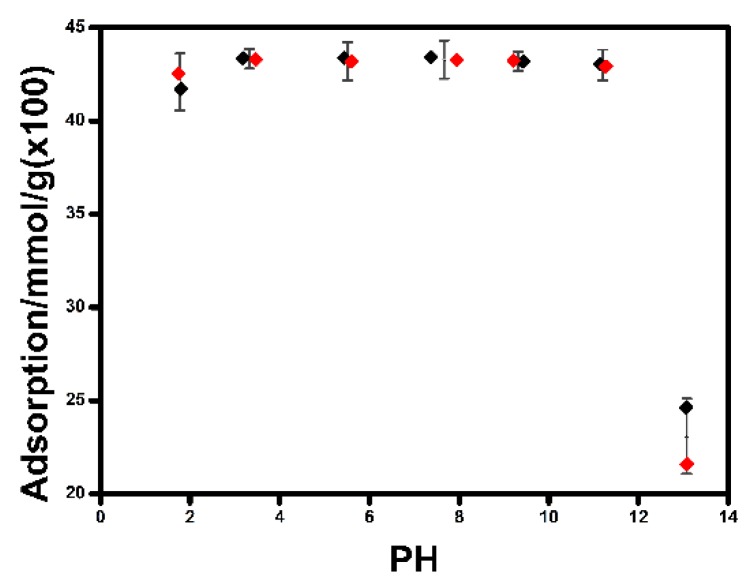
The adsorption of IL on sodium montmorillonite (Na-MMT) Na-MMT at different solution pH.

**Figure 5 molecules-24-01396-f005:**
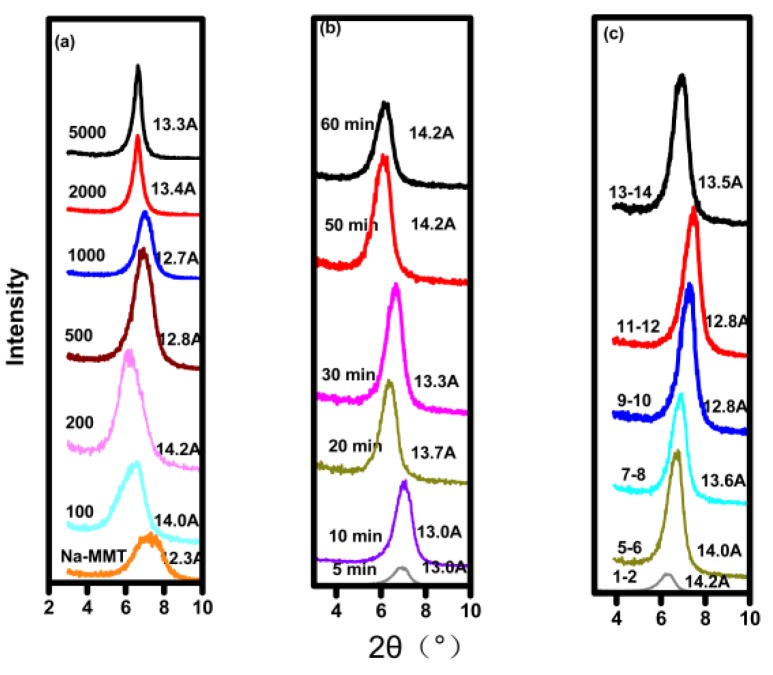
(**a**) X-ray diffraction(XRD) XRD patterns of raw Na-MMT and Na-MMT modified with different concentrations of IL; (**b**) XRD patterns of Na-MMT modified with 1000 mg/L IL solution for a different time; (**c**) XRD patterns of Na-MMT modified with 1000 mg/L IL solution at different solution pH.

**Figure 6 molecules-24-01396-f006:**
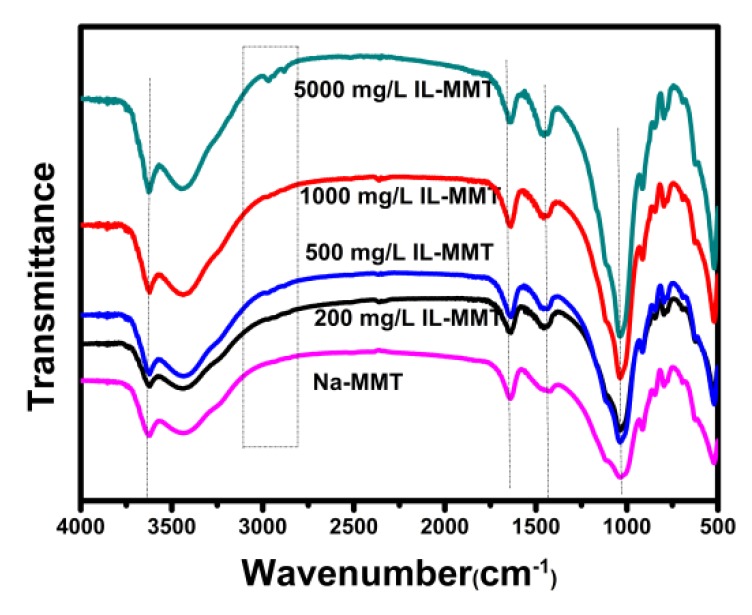
Fourier transform infrared(FTIR)spectra of IL, raw Na-MMT and Na-MMT modified with 1000 mg/L IL solution.

**Figure 7 molecules-24-01396-f007:**
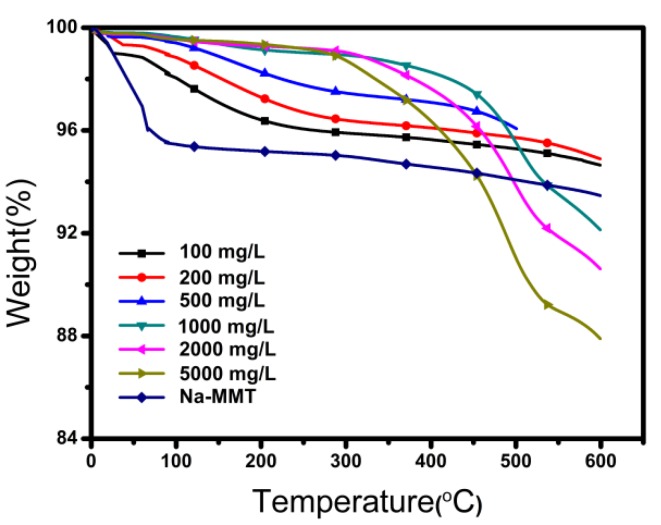
Thermogravimetric analyzer (TGA)TG patterns of raw Na-MMT and Na-MMT adsorbed with different concentrations of IL.

**Figure 8 molecules-24-01396-f008:**
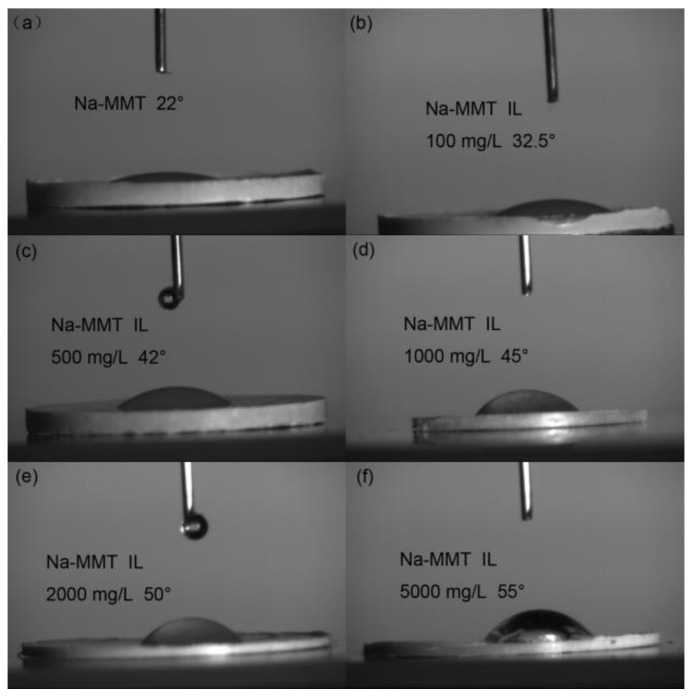
Contact angles of raw Na-MMT and modified Na-MMT with different concentrations of IL.

**Figure 9 molecules-24-01396-f009:**
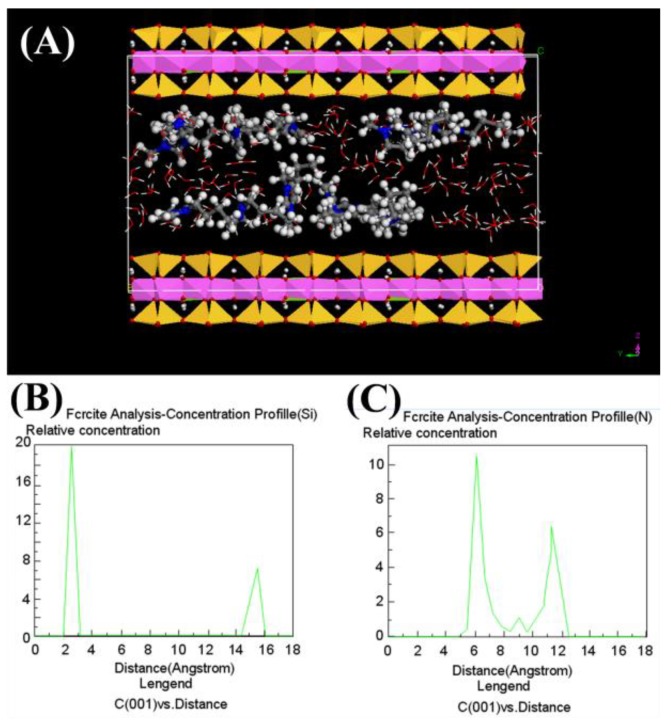
(**A**) Molecular simulation of IL conformation in the interlayer of Na-MMT; (**B**,**C**) the possible locations of two different N in the imidazole ring. In (A) blue is N, dark gray is C, and light gray is H.

**Figure 10 molecules-24-01396-f010:**
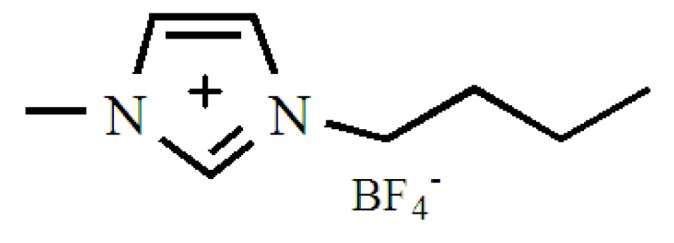
The IL formula.

**Figure 11 molecules-24-01396-f011:**
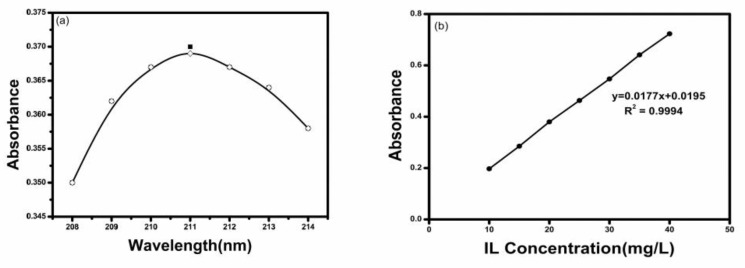
(**a**) the absorbance of the IL at different adsorption wavelength; (**b**) absorbance at 211 nm for IL solution with different concentrations and the fitted calibration curve.

**Table 1 molecules-24-01396-t001:** The kinetic parameters obtained by the pseudo-second-order model.

C_0_(mmol/L)	q _e(exp)_ (mmol/g)	q _e(cal)_ (mmol/g)	R^2^
4.42	0.43 ± 0.002	0.43	1.00

**Table 2 molecules-24-01396-t002:** The isotherm parameters obtained by fitting the experimental data to the Langmuir isotherm model.

q_m(cal)_ (mmol/g)	q_m(exp)_ (mmol/g)	K_b_ (L/mmol)	R^2^
0.83	0.70 ± 0.01	4.60	0.78

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
