# Peer review of "Study on the Adsorption Behavior between an Imidazolium Ionic Liquid and Na-Montmorillonite"

_molecules, 2019, doi:10.3390/molecules24071396_

Round 1
Reviewer 1 Report
The present manuscript reports sorption studies of ionic liquid on sodium montmorillonite. The author characterized using various techniques including XRD, TGA, UV-Vis. However, the paper is only a report of an experimental procedure, no significant or insights are presented. I’m sorry to say but I could not able to find the novelty from this manuscript. Nevertheless, please address the following comments and experiments before submitting to another journal
1) What is the novelty of this work compared to the previous work of ionic liquid sorption on Na MMT?
2) Explain briefly the use of IL in drilling fluids in the introduction section.
3) Author claims that adsorption behavior of the IL by Na-MMT has not reported? However, similar work using various ILs was reported in ref. 19
4) Please explain the specific reason to choose 1-butyl-3-methylimidazolium tetrafluoroborate ionic liquid? Is ILA only used in drilling fluids? BMImCl is already reported in the literature ref. 19. What significant impact this work has resulted by changing the anion?
5) 1-butyl-3-methylimidazolium tetrafluoroborate ionic liquid is abbreviated as ILA. It is not an appropriate abbreviation for the reported IL. Either use as IL or [BmIm][BF4]
6) The author mentions that ILA concentration were measured with UV-Vis spectrophotometer at 211 nm. Show the kinetics of the experiments and explain how exactly the absorbance of the absorbate increases? The increase could be background noise.
7) All the ILs absorbed within 5 min and sorption remains constant after prolong period of contact? What significant insights can be learnt from these results?
8) XRD data show that d (001) spacing increases with increase in ILA concentration to 200 mg/L then it remains constant until 1000 mg/L followed by an increase in spacing with an increase in concentration. The trend is not consistent, please clarify?
9) Further, the initial concentration of ILA at 1000 mg/L in figure 6a is 12.7 A°. Whereas in figure 6 b, 1000 mg/L IL varied with different time has different values? The results show some discrepancy. Please explain the protocol here carefully
Author Response
We greatly appreciate the suggestions of the referee. Our responses are as follows:
For comment 1
The referee provides us with a good suggestion. According to the suggestions of the referee, we revised the related parts of our manuscript (from Line 65 to Line 73) and made the novelty of the paper clear.
Previous work of ionic liquid sorption on Na-MMT mainly focus on the application of the IL-interacted Na-MMT at high temperature or the adsorption mechanism, few literature reported the microscopic structure of the interacted IL in the interlayer through the adsorption behavior. This work mainly investigated the microscopic mechanism of the IL by Na-MMT through the adsorption behavior.
For comment 2
According to the suggestions of the referee, we have made substantial revisions (from Line 67 to 69).
For comment 3
This comment is similar with comment 1. According to the suggestions of the referee, we have made substantial revisions from Line 65 to Line 73.
For comment 4
Ionic liquids (ILs) are organic salts which consist of cations and anions. One important property of ILs is its high thermal stability. The thermal stability of ILs is determined not only by the cations but the anions. In our previous work, we found that the thermal degrade temperature of the ionic liquid (1-butyl-3-methylimidazolium tetrafluoroborate: IL) was about 330℃, much more higher than that of 1-butyl-3-methylimidazolium chloride (BMIMCl). The thermal degrade temperature of the latter was about 240℃. Meanwhile, the thermal stability of the modified Na-MMT with the IL is much higher than that of the modified Na-MMT with BMIMCl. These two ionic liquids were added into water-based drilling fluids separately and these two ionic liquids can improve the properties of water-based drilling fluids at high temperature. But the IL can better improve the performance of the drilling fluids at high temperature compared to BMIMCl, which is required for the high-temperature resistant drilling fluids in deeper drilling. So we choose the IL as the additive of drilling fluids and investigated the adsorption behavior between the IL and Na-MMT.
For comment 5
According to the suggestions of the referee, we have made revisions in the relative part.
For comment 6
According to the suggestion of referee, we added the Fig. 2 (a) the absorbance of the IL at different adsorption wavelength (b) the calibration curve (From Line 103 to 106).
In our experiments, the calibration curve should be obtained to calculate the adsorbed amount of the IL. Firstly, the absorbance of the IL with low concentration at different adsorption wavelength was measured and the maximum adsorption wavelength can be obtained. The results can be seen the following figure (a). And then the calibration curve was obtained by measuring the absorbance values of 10, 20, 30, 40, 50 and 60 mg/L at the maximum adsorption wavelength. The results was shown in the following figure (b). As written in the revised manuscript the regression coefficient was greater than 0.999. Hence, the calibration curve (y=0.0177x+0.0195) can be used to calculate the adsorbed amount of the IL.
For comment 7
As written in our revised manuscript from Line 126 to Line 135, the adsorbed amount of the IL by Na-MMT at 5 min and 30 min was 0.429mmol/g and 0.436 mmol/g, respectively. Although the two values are very close, the adsorption of the IL by Na-MMT can reach the equilibrium when the maximal adsorbed amount of 0.436 mmol/g at 30 min. It can be seen from Fig.3(a) that the adsorbed amount of the IL by Na-MMT remain constant after 30 min. The most of the IL was adsorbed by Na-MMT within 5 min, the highest adsorption rate (r) of 98. 0 % was observed at the initial concentration of 1000 mg/L even if prolonging the period of contact`(Fig.4b). The rapid adsorption of the IL by Na-MMT suggested the ionic liquid had a great affinity for Na-MMT surfaces. This result also provides the basis for the application concentration of the IL.
For comment 8
The referee provides us with a good suggestion. According to the suggestions of the referee, we clarified some discussion in our manuscript (seen from Line 269 to Line 273) .
The d001-spacing of the modified Na-MMT did not increase with the increase in IL concentration of the IL. These results show that the d001-spacing of Na-MMT was not only dependent on the adsorption amount of IL but the arrangement of interacted cationic group in the interlayer of Na-MMT. Maybe the cationic group of the IL adopts vertical orientation which leads to the larger the d001-spacing of modified Na-MMT with lower concentration of the IL (100 mg/L). But when the concentration of the IL increases to higher, the cationic group of the IL adopts lying parallel orientation which leads to the smaller the d001-spacing of modified Na-MMT. Only one another literature reported similar this speculation.
For comment 9
We greatly appreciate the suggestion of the referee. According to the suggestions of the referee, we revised the related parts of our manuscript (from Line 276 to Line 283) and explain the results carefully.
The d001-spacing of the modified Na-MMTs is 12.7 Å as the initial concentration of IL was 1000 mg/L (Fig.7a). The modified Na-MMT samples with different initial concentration were prepared at the same adsorption time of 24 h. It was mentioned in the Line 93 to 95. The value of 12.7 Å is very close to 13.0 Å, the latter value of 13.0 Å is the d001-spacing of the modified Na-MMTs when the adsorption time was 5 min or 10 min. Although the results show some discrepancy, we think the results are reasonable. The d001-spacing of the modified Na-MMTs with 1000 mg/l IL varied with different time. As mentioned before, the d001-spacing of Na-MMT was not only dependent on the adsorption amount of IL but the arrangement of interacted cationic group in the interlayer of Na-MMT. For the modified Na-MMTs with IL of one same concentration, the d001-spacing of Na-MMT maybe not only dependent on the adsorption amount of IL but the adsorption time. The adsorption behavior is dynamic and the orientation adopted by the adsorbed cation group of IL should be varied with the different adsorption time. The adsorbed cation of IL maybe adopt a tilt at one time and at another time it maybe adopt vertical orientation in the interlayer of Na-MMT.
We greatly appreciate the suggestions of the referee. Our responses are as follows:
For comment 1
The referee provides us with a good suggestion. According to the suggestions of the referee, we revised the related parts of our manuscript (from Line 65 to Line 73) and made the novelty of the paper clear.
Previous work of ionic liquid sorption on Na-MMT mainly focus on the application of the IL-interacted Na-MMT at high temperature or the adsorption mechanism, few literature reported the microscopic structure of the interacted IL in the interlayer through the adsorption behavior. This work mainly investigated the microscopic mechanism of the IL by Na-MMT through the adsorption behavior.
For comment 2
According to the suggestions of the referee, we have made substantial revisions (from Line 67 to 69).
For comment 3
This comment is similar with comment 1. According to the suggestions of the referee, we have made substantial revisions from Line 65 to Line 73.
For comment 4
Ionic liquids (ILs) are organic salts which consist of cations and anions. One important property of ILs is its high thermal stability. The thermal stability of ILs is determined not only by the cations but the anions. In our previous work, we found that the thermal degrade temperature of the ionic liquid (1-butyl-3-methylimidazolium tetrafluoroborate: IL) was about 330℃, much more higher than that of 1-butyl-3-methylimidazolium chloride (BMIMCl). The thermal degrade temperature of the latter was about 240℃. Meanwhile, the thermal stability of the modified Na-MMT with the IL is much higher than that of the modified Na-MMT with BMIMCl. These two ionic liquids were added into water-based drilling fluids separately and these two ionic liquids can improve the properties of water-based drilling fluids at high temperature. But the IL can better improve the performance of the drilling fluids at high temperature compared to BMIMCl, which is required for the high-temperature resistant drilling fluids in deeper drilling. So we choose the IL as the additive of drilling fluids and investigated the adsorption behavior between the IL and Na-MMT.
For comment 5
According to the suggestions of the referee, we have made revisions in the relative part.
For comment 6
According to the suggestion of referee, we added the Fig. 2 (a) the absorbance of the IL at different adsorption wavelength (b) the calibration curve (From Line 103 to 106).
In our experiments, the calibration curve should be obtained to calculate the adsorbed amount of the IL. Firstly, the absorbance of the IL with low concentration at different adsorption wavelength was measured and the maximum adsorption wavelength can be obtained. The results can be seen the following figure (a). And then the calibration curve was obtained by measuring the absorbance values of 10, 20, 30, 40, 50 and 60 mg/L at the maximum adsorption wavelength. The results was shown in the following figure (b). As written in the revised manuscript the regression coefficient was greater than 0.999. Hence, the calibration curve (y=0.0177x+0.0195) can be used to calculate the adsorbed amount of the IL.
For comment 7
As written in our revised manuscript from Line 126 to Line 135, the adsorbed amount of the IL by Na-MMT at 5 min and 30 min was 0.429mmol/g and 0.436 mmol/g, respectively. Although the two values are very close, the adsorption of the IL by Na-MMT can reach the equilibrium when the maximal adsorbed amount of 0.436 mmol/g at 30 min. It can be seen from Fig.3(a) that the adsorbed amount of the IL by Na-MMT remain constant after 30 min. The most of the IL was adsorbed by Na-MMT within 5 min, the highest adsorption rate (r) of 98. 0 % was observed at the initial concentration of 1000 mg/L even if prolonging the period of contact`(Fig.4b). The rapid adsorption of the IL by Na-MMT suggested the ionic liquid had a great affinity for Na-MMT surfaces. This result also provides the basis for the application concentration of the IL.
For comment 8
The referee provides us with a good suggestion. According to the suggestions of the referee, we clarified some discussion in our manuscript (seen from Line 269 to Line 273) .
The d001-spacing of the modified Na-MMT did not increase with the increase in IL concentration of the IL. These results show that the d001-spacing of Na-MMT was not only dependent on the adsorption amount of IL but the arrangement of interacted cationic group in the interlayer of Na-MMT. Maybe the cationic group of the IL adopts vertical orientation which leads to the larger the d001-spacing of modified Na-MMT with lower concentration of the IL (100 mg/L). But when the concentration of the IL increases to higher, the cationic group of the IL adopts lying parallel orientation which leads to the smaller the d001-spacing of modified Na-MMT. Only one another literature reported similar this speculation.
For comment 9
We greatly appreciate the suggestion of the referee. According to the suggestions of the referee, we revised the related parts of our manuscript (from Line 276 to Line 283) and explain the results carefully.
The d001-spacing of the modified Na-MMTs is 12.7 Å as the initial concentration of IL was 1000 mg/L (Fig.7a). The modified Na-MMT samples with different initial concentration were prepared at the same adsorption time of 24 h. It was mentioned in the Line 93 to 95. The value of 12.7 Å is very close to 13.0 Å, the latter value of 13.0 Å is the d001-spacing of the modified Na-MMTs when the adsorption time was 5 min or 10 min. Although the results show some discrepancy, we think the results are reasonable. The d001-spacing of the modified Na-MMTs with 1000 mg/l IL varied with different time. As mentioned before, the d001-spacing of Na-MMT was not only dependent on the adsorption amount of IL but the arrangement of interacted cationic group in the interlayer of Na-MMT. For the modified Na-MMTs with IL of one same concentration, the d001-spacing of Na-MMT maybe not only dependent on the adsorption amount of IL but the adsorption time. The adsorption behavior is dynamic and the orientation adopted by the adsorbed cation group of IL should be varied with the different adsorption time. The adsorbed cation of IL maybe adopt a tilt at one time and at another time it maybe adopt vertical orientation in the interlayer of Na-MMT.
We greatly appreciate the suggestions of the referee. Our responses are as follows:
For comment 1
The referee provides us with a good suggestion. According to the suggestions of the referee, we revised the related parts of our manuscript (from Line 65 to Line 73) and made the novelty of the paper clear.
Previous work of ionic liquid sorption on Na-MMT mainly focus on the application of the IL-interacted Na-MMT at high temperature or the adsorption mechanism, few literature reported the microscopic structure of the interacted IL in the interlayer through the adsorption behavior. This work mainly investigated the microscopic mechanism of the IL by Na-MMT through the adsorption behavior.
For comment 2
According to the suggestions of the referee, we have made substantial revisions (from Line 67 to 69).
For comment 3
This comment is similar with comment 1. According to the suggestions of the referee, we have made substantial revisions from Line 65 to Line 73.
For comment 4
Ionic liquids (ILs) are organic salts which consist of cations and anions. One important property of ILs is its high thermal stability. The thermal stability of ILs is determined not only by the cations but the anions. In our previous work, we found that the thermal degrade temperature of the ionic liquid (1-butyl-3-methylimidazolium tetrafluoroborate: IL) was about 330℃, much more higher than that of 1-butyl-3-methylimidazolium chloride (BMIMCl). The thermal degrade temperature of the latter was about 240℃. Meanwhile, the thermal stability of the modified Na-MMT with the IL is much higher than that of the modified Na-MMT with BMIMCl. These two ionic liquids were added into water-based drilling fluids separately and these two ionic liquids can improve the properties of water-based drilling fluids at high temperature. But the IL can better improve the performance of the drilling fluids at high temperature compared to BMIMCl, which is required for the high-temperature resistant drilling fluids in deeper drilling. So we choose the IL as the additive of drilling fluids and investigated the adsorption behavior between the IL and Na-MMT.
For comment 5
According to the suggestions of the referee, we have made revisions in the relative part.
For comment 6
According to the suggestion of referee, we added the Fig. 2 (a) the absorbance of the IL at different adsorption wavelength (b) the calibration curve (From Line 103 to 106).
In our experiments, the calibration curve should be obtained to calculate the adsorbed amount of the IL. Firstly, the absorbance of the IL with low concentration at different adsorption wavelength was measured and the maximum adsorption wavelength can be obtained. The results can be seen the following figure (a). And then the calibration curve was obtained by measuring the absorbance values of 10, 20, 30, 40, 50 and 60 mg/L at the maximum adsorption wavelength. The results was shown in the following figure (b). As written in the revised manuscript the regression coefficient was greater than 0.999. Hence, the calibration curve (y=0.0177x+0.0195) can be used to calculate the adsorbed amount of the IL.
For comment 7
As written in our revised manuscript from Line 126 to Line 135, the adsorbed amount of the IL by Na-MMT at 5 min and 30 min was 0.429mmol/g and 0.436 mmol/g, respectively. Although the two values are very close, the adsorption of the IL by Na-MMT can reach the equilibrium when the maximal adsorbed amount of 0.436 mmol/g at 30 min. It can be seen from Fig.3(a) that the adsorbed amount of the IL by Na-MMT remain constant after 30 min. The most of the IL was adsorbed by Na-MMT within 5 min, the highest adsorption rate (r) of 98. 0 % was observed at the initial concentration of 1000 mg/L even if prolonging the period of contact`(Fig.4b). The rapid adsorption of the IL by Na-MMT suggested the ionic liquid had a great affinity for Na-MMT surfaces. This result also provides the basis for the application concentration of the IL.
For comment 8
The referee provides us with a good suggestion. According to the suggestions of the referee, we clarified some discussion in our manuscript (seen from Line 269 to Line 273) .
The d001-spacing of the modified Na-MMT did not increase with the increase in IL concentration of the IL. These results show that the d001-spacing of Na-MMT was not only dependent on the adsorption amount of IL but the arrangement of interacted cationic group in the interlayer of Na-MMT. Maybe the cationic group of the IL adopts vertical orientation which leads to the larger the d001-spacing of modified Na-MMT with lower concentration of the IL (100 mg/L). But when the concentration of the IL increases to higher, the cationic group of the IL adopts lying parallel orientation which leads to the smaller the d001-spacing of modified Na-MMT. Only one another literature reported similar this speculation.
For comment 9
We greatly appreciate the suggestion of the referee. According to the suggestions of the referee, we revised the related parts of our manuscript (from Line 276 to Line 283) and explain the results carefully.
The d001-spacing of the modified Na-MMTs is 12.7 Å as the initial concentration of IL was 1000 mg/L (Fig.7a). The modified Na-MMT samples with different initial concentration were prepared at the same adsorption time of 24 h. It was mentioned in the Line 93 to 95. The value of 12.7 Å is very close to 13.0 Å, the latter value of 13.0 Å is the d001-spacing of the modified Na-MMTs when the adsorption time was 5 min or 10 min. Although the results show some discrepancy, we think the results are reasonable. The d001-spacing of the modified Na-MMTs with 1000 mg/l IL varied with different time. As mentioned before, the d001-spacing of Na-MMT was not only dependent on the adsorption amount of IL but the arrangement of interacted cationic group in the interlayer of Na-MMT. For the modified Na-MMTs with IL of one same concentration, the d001-spacing of Na-MMT maybe not only dependent on the adsorption amount of IL but the adsorption time. The adsorption behavior is dynamic and the orientation adopted by the adsorbed cation group of IL should be varied with the different adsorption time. The adsorbed cation of IL maybe adopt a tilt at one time and at another time it maybe adopt vertical orientation in the interlayer of Na-MMT.

Reviewer 2 Report
In this work, the authors investigate the adsorption of ionic liquid on Na-montmorillonite. The present investigation fit with the journal topic and is quite interesting. However, I have several concerns about this work that have to be significantly improved before consideration for publication.
Major points.
About the kinetics : There are several inconsistence in the result show in figure 2. The values of qt/T (fig2b) do not correspond to the qt (fig2b). The calculations have to be verified in order to recalculate the kinetic. How does the qe(cal) was obtained ?
About the isotherm : The fit of Langmuir isotherm has to be shown. Kb is noted as Langmuir constant and Kl in equation 4. Please use the same letter.
In section 3.3 it appear that the clay have different cation. This is not consistent with the denomination Na-MMT. The chemical composition of clay has to be given. This has a real importance since the cation has an influence on the exchange.
At least one series of absorbance measured after IL adsorption has to be provided.
In all figure where are reported data the error bar should be added
The conclusion obtained from XRD, FTIR and molecular dynamic simulation should be supported by NMR experiments.
Minor points
The concentration in kinetic section is in mg/mg and in isotherm mmol/g. Please use the same unit.
The figue 2 fig 3 fig 5 is unreadable, The quality of figure have to be improved (size of typo and symbol). The caption has to give the signification of red and black symbols
The author should cite new paper on thid field : Colloids and Surfaces A: Physicochemical and Engineering Aspects 558, 219-227, Chemosphere 184, 1355-1361
In table 2 qe is noted two times
L42 “can’t disperse” should be “cannot disperse”
Author Response
We greatly appreciate the suggestions of the referee. Our responses are as follows:
For comment 1
According to the suggestion of the reviewer, we recalculated the data and the kinetic model was fitted the experimental data. The values of t/qt (Fig.3b) corresponded to the qt (Fig.3a). Because the unit of the qt in Fig. 3a is mg/g and the unit of the qt in Fig. 3b is mmol/g, it seemed to be inconsistence. According to the suggestion of the reviewer, we use the same unit (mmol/g) and revised the relative parts. We also added the calculation equation (from Line 143 to Line 146). In this work the pseudo-second-order model was obtained by the experimental data and the qe(cal) was obtained by the linear regression equation.
For comment 2
According to the suggestion of the reviewer, we use the same letter and revised the relative part in the Line 219.
For comment 3
We greatly appreciate the suggestion of the referee and gave us a chance to correct the mistake. We revised the relative part and added chemical formulas of the clay in the Experimental work in Line 86-88.
For comment 4
According to the suggestion of referee, we added the Fig. 2 (a) the absorbance of the IL at different adsorption wavelength (b) the calibration curve (From Line 104 to 106).
For comment 5
In the Fig. 2(a), Fig.3(a), Fig. 4(a), Fig.6 (a) the error bar had been added in our manuscript. According to the suggestion of referee, we revised the relative part the Experimental work (seen from Line 103 to Line 104).
For comment 6
This study is aimed to investigate the microscopic mechanism of the IL by Na-MMT through the adsorption behavior. X-ray diffraction is used to measure d001-spacing values of Na-MMT and the chemical composition of materials can be measured qualitatively through FTIR analyses. Whether the cation of IL insert into the interlayer of Na-MMT can be determined by the d001-spacing values of Na-MMT combined with FTIR analyses. We also speculated the orientation in light of the d001-spacing values of Na-MMT. Meantime, the atomic arrangement of IL were simulated by the software. It is better to support the conclusion by adding the NMR experiments based on the above discussion. We will do more work in our future study.
For comment 7
According to the suggestion of the reviewer, we use the same unit and revised the relative parts.
For comment 8
We greatly appreciate the suggestion of the referee. We improved the quality and resolution of all the figures in the manuscript according to the standard of the publication.
For comment 9
According to the suggestion of the referee, we cited a new paper on this field and revised the relative part in Line 458-461.
For comment 10
qe is noted two times in Table 2. One qe is obtained by the expeimental data, another qe is obtained by a linear regression.
For comment 11
According to the suggestion of the referee, we revised the relative part in Line 42.

Round 2
Reviewer 1 Report
In the author response document, the author has addressed few questions to the point, however the authors failed to consider other comments into account. Please carefully read the suggestions and address them carefully for publication. In addition, please include the question while responding to it. It makes the reviewers life easier to understand.
1. According the previous literature (as author mentioned in the manuscript ref. 18, 19, 20), is adsorption or exchange of cation taking place between the clay and the ionic liquid?
2. The author also showed the desorbed cations and absorption of IL in Fig. 5. Sodium is good exchangeable cation? Please provide the reason based on the size of the inorganic cations? How is the experimental conducted?
3. In continuation… I assume the ionic liquid is exchanging with clay cation than absorbing?
4. According the experimental work, “the 0.2 g of clay was dispersed in 20 mL of IL…. Finally the filtered and precipitates were dried for measurement.” What temperature was used to dry the precipitates and how the author confirmed the excess amount of IL was washed out?
5. Please clearly explain the pH studies and how it is conducted?
6. Please provide the entire absorption spectra of the absorbance of the clay and then show the maximum absorbance inset.
7. Also, please provide how the measurements were conducted? Did the author dispersed precipitates in solution or solid-state measurements? It was not clear
8. Most of the figures are blurry and hard to visualize. Also, include error bars in the figures?
9. The thermal degradation temperature changes with adsorbed IL? Good but please provide the experimental work how the TGA is measured? After filtering, the excess amount of IL was not washed? It could be both adsorption and absorption on the clay
10. Line 269 to 272: Provide references and suggest that these parallel orientations is confirmed with simulation work provided in the manuscript
Author Response
We greatly appreciate the suggestions of the referee. In the light of the comments made by the editor and referee, we have revised our manuscript accordingly.
For comment 1
1. According the previous literature (as author mentioned in the manuscript ref. 18, 19, 20), is adsorption or exchange of cation taking place between the clay and the ionic liquid?
Generally, the adsorption mechanism of electrolyte by the solids can be classified into ion-exchange or ion-selective mechanism. That is to say, there have two kinds of adsorption mechanism of electrolyte by the solids: ion-exchange and ion-selective adsorption. The clay with negative charge is balanced by the cationic counter-ion. The cation of the clay can exchange with other cation in the electrolyte solution (cation-exchange adsorption). This conclusion is provided in the reference 19 and some books (Caenn R., Darley HCH., Gray GR. 2011. Composition and Properties of Drilling and Completion Fluids:152, Gulf Professional Publishing; Shen Z., Zhao ZG., Wang GT., 2012. Colloid and Surface Chemistry: 58, Chemical Industry Press; Zhu BY., Zhao ZG., 1996. Interface Chemistry: 25, Chemical Industry Press).
The reference 18 mainly focuses on the application of other ionic liquids in polymers and the exchanges of cation between the clay and other ionic liquids have been reported in the reference 19. The reference 20 mainly focuses on the application of the ionic liquid in drilling fluids.
For comments 2
2. The author also showed the desorbed cations and absorption of IL in Fig. 5. Sodium is good exchangeable cation? Please provide the reason based on the size of the inorganic cations? How is the experimental conducted?
As mentioned in the comment 1, the adsorption of electrolyte by the clay can be explained based on cation-exchange mechanism. When the clay adsorbs a kind of cation in the electrolyte, the equivalent cation of the clay should be exchanged. The ion-exchange adsorption follows the weigh action law. The exchange coefficient is related to the characteristics of the exchangeable cation. The higher ion valence and lower hydrated ionic radius (higher ionic radius) result in higher exchange coefficient, which has been reported in some books (Caenn R., Darley HCH., Gray GR. 2011. Composition and Properties of Drilling and Completion Fluids:152, Gulf Professional Publishing; Shen Z., Zhao ZG., Wang GT., 2012. Colloid and Surface Chemistry, Chemical Industry Press; 258; Zhu BY., Zhao ZG., 1996. Interface Chemistry, 257). For different ions with the same valence and solution concentration, the smaller ionic radius the ion is, the higher hydrated ionic radius the ion is, and consequently, the weaker exchange adsorption the ion is. The ionic radius of Na+ is relatively lower and thus sodium ions are good exchangeable ions.
In the revised manuscript, we deleted the sentence “the testing procedure was conducted according to the reported literature”, and added the revelant content in Line 122 to 126. And we also revised the experimental part in Line 92 to 102. “0.2 g of Na-MMT was dispersed in 20 mL of IL solution and the suspensions were shaken for 24 h at 30 °C. After being centrifuged at 7600 rpm for 30 min, the suspensions were filtered through 0.45 μm syringe filters, and then the filter liquor was analyzed for the concentrations of IL and the desorbed exchangeable cations. Each precipitate was washed with distilled water at least three times until no IL could be detected in the supernatant by a UV-Visible spectrophotometer (Model T6, China). The clay samples were obtained by drying the precipitates at 105°C for 24 h in an oven before measurement. The IL concentration was fixed at 1000 mg/L for the study of adsorption kinetics and pH effects on the adsorption of IL by Na-MMT. The pH value of the Na-MMT/IL dispersion was adjusted to about 2, 3, 5, 7, 9, 11 and 13 with HCl or KOH.”
For comment 3
3. In continuation… I assume the ionic liquid is exchanging with clay cation than absorbing?
As discussed above (reply to comment 1), the adsorption mechanism of electrolyte by the solids can be classified into ion-exchange or ion-selective mechanism. The cation of the clay can exchange with other cation in the electrolyte solution (cation-exchange adsorption). For ionic liquids, the intercalation of cation groups of the IL into the clay can expand the spacing of the clay layers as proven by XRD, FTIR analyses. The intercalation confirmed that the adsorption of the IL through interlayer cation exchange mechanism. Therefore, it is reasonable to say that the ionic liquid is adsorbed by the Na-MMT, and the adsorption is through a cation-exchange mechanism.
For comment 4
4. According the experimental work, “the 0.2 g of clay was dispersed in 20 mL of IL…. Finally the filtered and precipitates were dried for measurement.” What temperature was used to dry the precipitates and how the author confirmed the excess amount of IL was washed out?
In our experiment, the dry clay samples were obtained by keeping the precipitates at 105°C for 24 h in an oven before measurement.
The adsorption of the IL by Na-MMT is mainly through cation-exchange mechanism, and the excess amount of IL is soluble in the filter liquor because IL is water-soluble(seen from Line . In order to make sure that the excess amount of IL was washed out, each precipitate was washed with distilled water at least three times until no IL could be detected in the supernatant by a UV-Visible spectrophotometer (Model T6, China). According to the experimental results, there were almost no excess “free” IL in the Na-MMT sample. The amount of IL in the supernatant is negligible even in the first washing procedure.
According to the suggestion of the referee, we revised the relative part and added the preparation condition for the dry clay sample “the precipitates were dried at 105°C for 24 h in an oven before measurement” in the revised manuscript to make it clear. We also added the water solubility of IL in Line 83 to 84..
The revised part in Line 96 to Line 99: “Each precipitate was washed with distilled water at least three times until no IL could be detected in the supernatant by a UV-Visible spectrophotometer (Model T6, China). The clay samples were obtained by drying the precipitates at 105°C for 24 h in an oven before measurement.”.
For comment 5
5. Please clearly explain the pH studies and how it is conducted?
The IL concentration was fixed at 1000 mg/L for the study of the effect of solution pH on the adsorption of IL by Na-MMT. After the Na-MMTs were added into the IL solutions, the pH of the Na-MMT/IL dispersion was adjusted at about 2, 3, 5, 7, 9, 11 and 13 with HCl or KOH. Because of the electrostatic interaction, the Na-MMT with negative charge adsorbed the cation (Na+ or Ca2+), as described in Line 35 in the revised manuscript. The Na-MMT exchanges with the cation in the solution and the anion diffusely distributed in the solution, which is called diffuse double layer (Caenn R., Darley HCH., Gray GR. 2011. Composition and Properties of Drilling and Completion Fluids:156, Gulf Professional Publishing). When the pH value of the solution increased to 13, the strong base solution may result in the lower zeta potential (ζ) of Na-MMT, which has been reported by some book (Shen Z., Zhao ZG., Wang GT., 2012. Colloid and Surface Chemistry, Chemical Industry Press; 258). Then, less cations may insert into the interlayer of Na-MMT with lower zeta potential (ζ). That is to say, the less amount of IL may be adsorbed by Na-MMT. According to the suggestion of the referee, we revised the relive part and made the explanation more clear (from Line 259-Line 262). “When the pH value of the dispersion increased to 13, the strong base solution may result in lower zeta potential of Na-MMT. Then less cations may insert into the interlayer of Na-MMT with lower zeta potential. That is to say, less amount of IL may be adsorbed by Na-MMT in solution with high pH value.”
For comment 6
6. Please provide the entire absorption spectra of the absorbance of the clay and then show the maximum absorbance inset.
The adsorbed amount of IL is calculated based on the difference between the final and initial IL concentration. Hence, only the UV-Vis spectrum of the IL is required, we provide the absorbance of the IL at different adsorption wavelength and show the maximum absorbance wavelength in the revised manuscript in Fig.2.
For comment 7
7. Also, please provide how the measurements were conducted? Did the author dispersed precipitates in solution or solid-state measurements? It was not clear
The amount of IL adsorbed by Na-MMT and the desorbed exchangeable cations were determined by measuring the concentration of IL and exchangeable cations in the filter liquor, rather than analyzing the modified clay samples after cation exchange. The precipitates will not be dispersed in solution again, and the precipitates were dried at 105°C for 24 h for further measurements (XRD, FT-IR, TGA and Contact angle).
As described in Line 92 to Line 96, the filter liquor was obtained by the following process: “0.2 g of Na-MMT was dispersed in 20 mL of IL solution and the suspensions were shaken for 24 h at 30 °C. After being centrifuged at 7600 rpm for 30 min, the suspensions were filtered through 0.45 μm syringe filters, and then the filter liquor was analyzed for the concentrations of IL and the desorbed exchangeable cations.”
As described in Line 103 to Line 109, the concentration of IL in the filter liquor was determined by UV-Vis spectrum: “The IL concentrations were measured with UV-Vis spectrophotometer at the wavelength of 211 nm [21], corresponding to its maximum absorbance wavelength. The calibration curve was obtained by measuring the absorbance values of the IL solution with varying concentration (10, 20, 30, 40, 50 and 60 mg/L) and the regression coefficient was greater than 0.999 (Fig.2). The adsorbed amount of IL can be calculated by the difference between the final and initial IL concentration.”
As described in Line 121 to Line 126, the concentration of desorbed exchangeable cations in the filter liquor was determined by Atomic Absorption Spectrum: “The desorbed exchangeable cations were measured by atomic adsorption on a Perkin Elmer Atomic Absorption Spectrometer (AAnalyst-100, USA). Seven standard solutions with concentrations from 0.2 to 3.0 mg/L for K+, Na+, and Mg2+, and from 1.0 to 25.0 mg/L for Ca2+ were used to make the calibration curve. The detection limit was 0.006, 0.01, 0.06, and 0.4 mg/L for Na+, K+, Mg2+ and Ca2+ with detection wavelengths at 589.0, 766.5, 285.2, and 422.7 nm, respectively.”
According to the suggestion of the referee, we revised the manuscript accordingly and added the part in Line 109 to Line 116 in the revised manuscript. “The equation is as follows:
(1)
(2)
Where qt and qe are the corresponding adsorbed amount of IL by Na-MMT at t and equilibrium time, respectively (mmol/g). Ct and Ce are the corresponding concentration of the IL at t and equilibrium time, respectively (mmol/L). C0 is the initial concentration of the IL (mmol/L), V is the volume of IL solution (L), M is the weight of adsorbent (g).”
We also revised the relative parts in the revised manuscript from Line 95 to Line 99 to make it clear. “and then the filter liquor was analyzed for the concentrations of IL and the desorbed exchangeable cations. Each precipitate was washed with distilled water at least three times until no IL could be detected in the supernatant by a UV-Visible spectrophotometer (Model T6, China). The clay samples were obtained by drying the precipitates at 105°C for 24 h in an oven before measurement.”
For comment 8
8. Most of the figures are blurry and hard to visualize. Also, include error bars in the figures?
According to the suggestion of the referee, we improved the quality and resolution of the figures, and added the error bar in some figures.
For comment 9
9. The thermal degradation temperature changes with adsorbed IL? Good but please provide the experimental work how the TGA is measured? After filtering, the excess amount of IL was not washed? It could be both adsorption and absorption on the clay
In our present work, TGA measurement was utilized to the thermal stability of the raw Na-MMT and Na-MMT modified with IL. The temperature range of the TGA measurement was from room temperature to 600°C, operating at a heating rate of 10 °C/min in an air atmosphere. According to the suggestion of the referee, we revised the relative part from Line 98 to 99. and We provided the parameters for the TGA measurement in Line 131 to Line 134. “The thermal stability of the clay samples was measured by the TA thermogravimetric analyzer (TGA) (SDT Q600, USA) from room temperature to 600°C, operating at a heating rate of 10 °C/min in an air atmosphere. ”
Furthermore, after filtering, the excess amount of IL was washed thoroughly with water. According to the suggestion of the referee, we revised the manuscript accordingly and added the purifying process for the Na-MMT modified with IL. As described in Line 96 to Line 98: “Each precipitate was washed with distilled water at least three times until no IL could be detected in the supernatant by a UV-Visible spectrophotometer (Model T6, China).”
According to the experimental results, there were almost no excess “free” IL in the Na-MMT sample (residual IL on the clay). The amount of IL in the supernatant is negligible even in the first washing procedure.
For comment 10
10. Line 269 to 272: Provide references and suggest that these parallel orientations is confirmed with simulation work provided in the manuscript
According to the suggestion of the referee, we added the basis for the speculation from Line 281 to Line 284: “ The d001-spacing of raw Na-MMT with one layer of water is 12.3 Å. The interlayer distance can be obtained by subtracting the thickness of dehydrated Na-MMT layer (9.6 Å) [24] from the observed d-spacing of the modified Na-MMT with IL. But the height of imidazole group with alkyl chain was about 3.3 Å [29].”
We also provided the reference (Takahashi C, Takashi S, Masayoshi F.Material Chemistry and Physics 2012; 135: 681–686).

Reviewer 2 Report
In this version only several correction was taken into account,
The figure 2, 3, 6 and 4 are still unreadable.
The error bar are not represented on the figure 3, 6 and 4 as well as in table 1 and 2
There are many mistake on the references list Please check name and surname carrefully ref 12, 18,, 19,26, 28, 30
Author Response
COMMENTS FROM EDITOR AND/OR REVIEWERS
Reviewer #2:
In this version only several correction was taken into account,
The figure 2, 3, 6 and 4 are still unreadable.
The error bar are not represented on the figure 3, 6 and 4 as well as in table 1 and 2
There are many mistake on the references list Please check name and surname carrefully ref 12, 18,, 19,26, 28, 30
For comment 1
The figure 2, 3, 6 and 4 are still unreadable.
The error bar are not represented on the figure 3, 6 and 4 as well as in table 1 and 2
According to the suggestion of the referee, we improved the quality of all the figures, and made it readable. We also added the error bar in the figure 3, 4 and 6 as well as table1 and 2 (seen in Line 170 and Line 222).
For comment 2
There are many mistake on the references list Please check name and surname carrefully ref 12, 18,, 19,26, 28, 30
We greatly appreciate the suggestion of the referee and and gave us a chance to correct the mistake. We have made substantial revisions (seen from Line 433 to Line 434, and Line 448 to Line 451, Line452 to Line 453,Line 472 to Line 474, Line 478 to Line 479,Line 485 to Line 486).
